# Hepatocellular Carcinoma: The Evolving Role of Systemic Therapies as a Bridging Treatment to Liver Transplantation

**DOI:** 10.3390/cancers16112081

**Published:** 2024-05-30

**Authors:** Yacob Saleh, Taher Abu Hejleh, Maen Abdelrahim, Ali Shamseddine, Laudy Chehade, Tala Alawabdeh, Issa Mohamad, Mohammad Sammour, Rim Turfa

**Affiliations:** 1Department of Internal Medicine, King Hussein Cancer Center, Amman 11941, Jordan; ta.11703@khcc.jo (T.A.H.); ta.11388@khcc.jo (T.A.); ms.14475@khcc.jo (M.S.); 2Section of GI Oncology, Houston Methodist Neal Cancer Center, Houston, TX 77030, USA; mabdelrahim@houstonmethodist.org; 3Division of Hematology and Oncology, Department of Internal Medicine, Naef K. Basile Cancer Institute, American University of Beirut Medical Center, Beirut P.O. Box 11-0236, Lebanon; as04@aub.edu.lb (A.S.); lc30@aub.edu.lb (L.C.); 4Department of Radiation Oncology, King Hussein Cancer Center, Amman 11941, Jordan; imohamad@khcc.jo

**Keywords:** hepatocellular carcinoma (HCC), systemic therapies, bridging, liver transplantation (LT), tyrosine kinase inhibitors (TKIs), immune checkpoint inhibitors (ICIs)

## Abstract

**Simple Summary:**

Hepatocellular carcinoma (HCC) is a common cancer and a leading cause of cancer-related deaths worldwide. However, HCC can be effectively treated in selected cases, with liver transplantation representing one of the limited options for potential cure. Unfortunately, many patients are ineligible for liver transplantation either due to an advanced tumor at initial diagnosis or due to disease progression while awaiting liver transplantation. Our review discusses the role of systemic therapies as a bridging treatment to liver transplantation, thereby enabling more HCC patients to undergo potentially curative liver transplantation.

**Abstract:**

Hepatocellular carcinoma (HCC) is the third most common cause of cancer-related deaths. Classically, liver transplantation (LT) can be curative for HCC tumors within the Milan criteria. Bridging strategies to reduce the dropouts from LT waiting lists and/or to downstage patients who are beyond the Milan criteria are widely utilized. We conducted a literature-based review to evaluate the role of systemic therapies as a bridging treatment to liver transplantation (LT) in HCC patients. Tyrosine kinase inhibitors (TKIs) can be used as a systemic bridging therapy to LT in patients with contraindications for locoregional liver-directed therapies. Immune checkpoint inhibitor (ICI) treatment can be utilized either as a monotherapy or as a combination therapy with bevacizumab or TKIs prior to LT. Acute rejection after liver transplantation is a concern in the context of ICI treatment. Thus, a safe ICI washout period before LT and cautious post-LT immunosuppression strategies are required to reduce post-LT rejections and to optimize clinical outcomes. Nevertheless, prospective clinical trials are needed to establish definitive conclusions about the utility of systemic therapy as a bridging modality prior to LT in HCC patients.

## 1. Introduction

Liver cancer is the sixth most commonly diagnosed cancer and the third most common cause of cancer-related deaths worldwide [1]. Hepatocellular carcinoma (HCC) accounts for approximately 80% of liver cancers, most often against the background of liver cirrhosis secondary to multiple risk factors, including hepatitis C virus (HCV) infection, hepatitis B virus (HBV) infection, alcohol-related liver disease (ALD) or metabolic dysfunction-associated steatotic liver disease (MASLD) [2,3].

Liver transplantation (LT) is the treatment of choice for HCC patients in the early stages if the tumor is unresectable [4,5]. There are various criteria used to assess eligibility for LT in HCC. Classically, the Milan criteria are utilized to define eligibility for LT, with the goal of achieving 4-year overall survival (OS) of more than 75%. The Milan criteria are included in the Barcelona Clinic Liver Cancer (BCLC) staging system, the American Association for the Study of Liver Diseases (AASLD) and the European Association for the Study of the Liver-European Organization for Research and Treatment of Cancer (EASL-EORTC) guidelines [5,6]. The University of California-San Francisco (UCSF) criteria represent another important expanded set of criteria for LT in HCC patients with comparable outcomes to the Milan criteria. The UCSF criteria are adopted in countries such as Australia and New Zealand [7,8,9]. Moreover, the United Network of Organ Sharing (UNOS) Down-Staging Criteria for LT are widely adopted in the USA, with a successful 5-year post-LT OS rate of 74% [10].

The biggest challenge for LT is organ shortage and long LT waiting lists, leading to patients’ dropout due to disease progression or liver decompensation [11]. In the USA, the national dropout rate from the transplant list for HCC patients reaches up to 29% as per the UNOS figures [12]. Thus, there is a need to implement therapies that control HCC until the availability of liver transplantation. This “bridging therapy” approach to transplantation aims either to reduce the number of dropouts from LT waiting lists for patients within the Milan criteria and/or to downstage patients who are beyond the Milan criteria [13]. Bridging strategies are either locoregional liver-directed or systemic therapies. Liver-directed therapies include radiofrequency ablation (RFA), microwave ablation (MWA), transarterial chemoembolization (TACE), transarterial radioembolization (TARE) and stereotactic body radiotherapy (SBRT) [14,15].

This review will focus on systemic therapies, including tyrosine kinase inhibitors (TKIs) and immune checkpoint inhibitors (ICIs), as bridging strategies to LT in HCC patients (Figure 1).

## 2. Systemic Therapies

### 2.1. Tyrosine Kinase Inhibitors (TKIs)

The therapeutic effect of TKIs is attained by binding to various tyrosine kinase receptors, thereby inhibiting downstream intracellular signaling, eventually resulting in apoptosis and anti-angiogenic effects, which play a major role in the tumor microenvironment of HCC [16]. Sorafenib, a small molecule TKI, was approved by the FDA for the treatment of advanced HCC based on the breakthrough results of the SHARP Trial in 2008 [17].

After approval for advanced HCC, TKIs were investigated as bridging treatments to LT. In 2010, Saidi et al. reported a case series of seven HCC patients whose tumors met the Milan criteria. All the patients were treated with sorafenib and six of them successfully underwent LT without local recurrence or distant metastasis [18]. In 2013, Vitale et al. reported another case series of six HCC patients who had Child–Pugh class A and intermediate stage disease. These patients received sorafenib before LT and the 4 patients who received sorafenib for a period of ≥2 months before LT were disease-free 27 to 41 months after LT [19]. In 2018, Golse et al. reported a cases series of five HCC patients, three of whom underwent hepatectomy or TACE and then received sorafenib as a bridging therapy before LT, while the other two patients received sorafenib as a downstaging therapy before LT. There were no tumor recurrences after LT within a 27-month follow-up [20]. In 2022, an observational study from France found that 62 out of 327 HCC patients listed for LT were treated with sorafenib; 50% of these patients received sorafenib because of HCC progression after locoregional therapy (LRT), the other 50% received sorafenib because of ineligibility to receive LRT. A total of 26 patients could progress to LT, where the 5-year OS and RFS were 77% and 48%, respectively [21].

Combination therapy with LRT and sorafenib as a bridging strategy to improve outcomes was also investigated. In 2014, 20 HCC patients who met the UCSF criteria for LT were randomized in a prospective trial to Y90 radioembolization with or without sorafenib (1:1 randomization ratio). Of the 20 patients, 17 underwent LT. The survival rates at 3 years were similar between the two groups. Combination therapy was associated with higher peri-transplant biliary complications and a trend toward more acute rejections [22]. In 2015, another prospective trial was published on 50 HCC patients who met the Milan criteria for LT and were randomized to TACE plus either sorafenib or placebo. A total of 17 patients could proceed to LT, and both groups had similar results in terms of the time to progression (TTP), tumor response and time to LT [23]. In 2022, a retrospective study on 128 HCC patients, whose tumors were either within or beyond the Milan criteria, found that those patients who received TACE plus sorafenib before LT achieved significantly better 5-year DFS compared to those who received TACE only; however, there was no significant difference in the 5-year OS rates between the two groups (77.8% vs. 61.5%, *p* value: 0.51) [24].

Cabozantinib is another TKI that is approved as a subsequent line of treatment for advanced HCC patients [25]. In 2022, Bhardwaj et al. reported two HCC patients who received cabozantinib after prior treatments with sorafenib and LRT. Both patients managed to progress to LT. The first patient developed disease recurrence 5 months after LT and the other patient was disease-free at the 21-month follow-up post LT [26].

In the Reflect trial, lenvatinib, a multi-kinase inhibitor, was compared to sorafenib in the first-line treatment of advanced HCC patients. Lenvatinib was non-inferior to sorafenib for survival. Lenvatinib achieved higher responses rates and better time to progression than sorafenib [27]. The effectiveness of lenvatinib compared to sorafenib was further confirmed in a meta-analysis [28]. Therefore, further exploration of lenvatinib in the bridging setting preceding LT can be promising.

#### Summary and Recommendations

The utilization of TKIs as a bridging treatment to LT in HCC patients with contraindications for locoregional liver-directed therapies is reported in the literature (Table 1). Most published reports examined sorafenib and found encouraging survival rates. However, prospective clinical trials are needed to establish definitive conclusions about the utility of TKI monotherapy as a bridging modality prior to LT in HCC patients.

### 2.2. Immune Checkpoint Inhibitors (ICIs)

#### 2.2.1. Efficacy and Safety

Over the past few years, immune checkpoint inhibitors (ICIs) have revolutionized the field of oncology, leading to prolonged survival with manageable side effects in various cancers [29]. In 2020, the combination therapy of atezolizumab (ICI) with bevacizumab was granted FDA approval for the first-line treatment of advanced HCC patients based on the superior survival outcome compared to the classical treatment with sorafenib [30].

The mechanism of action of ICIs involves activation of the suppressed innate immune system, which is contradictory to the functions of the immunosuppressant agents that are typically used post-LT. Consequently, the use of ICIs may increase the risk of rejection of the transplanted liver, a formidable potential consequence of using ICIs after liver transplantation. In a retrospective study by Wang et al., 16 HCC patients received ICI treatment before LT, and the study showed that the median time of acute rejection was 7 days post-LT and a shorter time interval (TI) between the last dose of ICI and LT increased the risk of postoperative rejection (median TI: 21 days vs. 60 days, *p* = 0.01); however, there were no immune-related graft losses [31]. Therefore, it is essential to establish a safe washout period, which is defined as the period between the last ICI dose and LT. In 2023, Kuo et al. concluded in a retrospective study that 42 days is a safe washout period for bridging ICI therapy with either atezolizumab, nivolumab or pembrolizumab before LT [32].

Data on the efficacy and safety of ICIs as bridging treatment to LT are accumulating. In 2021, Tabrizian et al. reported a single-center case series of nine HCC patients whose tumors met either the Milan or UCSF criteria. All the patients received nivolumab ICI therapy before being successfully bridged to liver transplant. Interestingly, 80% of the patients underwent LT within 4 weeks of the last dose of nivolumab and there were no instances of severe allograft rejections, tumor recurrences, or deaths at a median follow-up of 16 months after LT [33]. Moreover, in 2021, Chen et al. reported a case series of five HCC patients whose tumors were beyond the Milan Criteria. They all received bridging/downstaging treatment with nivolumab, the mean washout period was 63.8 days, and none of the patients developed biopsy-proven acute rejection (BPAR); however, two of them had HCC recurrences on follow-up [34]. In 2022, Schnickel et al. reported a case series of five HCC patients who received nivolumab ICI therapy before LT. The 2 patients who underwent LT within 3 months from the last dose of nivolumab developed BPAR and severe hepatic necrosis; however, BPAR was observed in none of the patients who underwent LT >3 months from the last dose of nivolumab [35]. Multiple case reports are published on ICI monotherapy as a bridging treatment to LT [36,37,38,39].

Combination therapies of ICIs with TKIs were also investigated as a bridging treatment strategy to LT. In 2021, Qiao et al. reported a cohort of seven HCC patients who received lenvatinib in combination with either pembrolizumab or camrelizumab ICI therapy prior to LT. The biopsy-proven acute rejection rate was 14.3% [40]. In 2022, Abdelrahim et al. published a case report on an HCC patient whose tumor was beyond the Milan criteria. The patient received atezolizumab plus bevacizumab combination therapy prior to successfully progressing to LT, with no HCC recurrence after 12 months of follow-up [41]. In 2023, Schmiderer et al. published another case report on an HCC patient who was successfully transplanted after receiving the atezolizumab plus bevacizumab combination therapy [42].

##### Summary and Recommendations

There is growing evidence in the literature regarding the role of ICIs as bridging treatment to LT in HCC. ICI treatment can be utilized either as a monotherapy or as a combination therapy with bevacizumab or TKIs prior to LT (Table 2). Acute rejection after LT is a concern in the context of ICI treatment. Thus, a safe ICI washout period before LT and cautious post-LT immunosuppression strategies are required to reduce post-LT rejections and to optimize clinical outcomes [31,32]. Furthermore, prospective clinical trials are needed to establish definitive conclusions about the utility of ICIs as a bridging modality prior to LT in HCC patients.

#### 2.2.2. Response Assessment

The classic radiologic disease assessment criteria, such as the response evaluation criteria in solid tumors (RECIST), may not adequately evaluate the response in HCC. These criteria rely on the tumor size, which can remain unchanged in locally or systemically treated HCC due to multiple factors, such as treatment-induced necrosis, the presence of ascites and reactive lymph nodes [44]. This results in an underestimation of the HCC tumor response. Therefore, in 2001, the European Association for the Study of the Liver (EASL) criteria were put forward to evaluate viable lesions on abdominal magnetic resonance imaging (MRI) [45]. Using this method, the arterially enhancing tumor burden is calculated in two dimensions. In 2008, the RECIST criteria were modified to the mRECIST, which include changes in tumor arterial enhancement [46]. The mRECIST can be applied to contrast-enhanced, multiphasic computed tomography (CT) or MRI. One of the advantages over the EASL criteria is that mRECIST provides recommendations for new lesions and non-target lesion selection, such as portal vein thrombosis, lymph node at the porta hepatis, ascites or pleural effusion [47]. The objective response rate was evaluated as a surrogate endpoint for overall survival (OS) in a systematic review of 14,056 patients with HCC treated with ICIs. The results of the meta-analysis showed that the objective response predicted by mRECIST (OR-mRECIST) was an independent predictor of OS and that the OR-mRECIST correlated better with the OS than the RECIST [48].

With the addition of immunotherapy to the armamentarium of treatment for HCC, pseudo-progression on imaging became a concern but was dismissed as it is rarely seen in HCC [49]. Therefore the immune response evaluation criteria in solid tumors (iRECIST), as designed to assess pseudo-progression, may not be required in HCC treated with immunotherapy [49]. On the contrary, tumor shrinkage is seen with immune checkpoint inhibitors and thus the RECIST criteria can be applied in this setting, as was the case in several clinical trials on immunotherapy in HCC.

##### Summary and Recommendations

Both the RECIST and mRECIST criteria (Table 3) are recommended by the guidelines for disease evaluation after systemic treatment [50].

#### 2.2.3. Biomarkers

Companion diagnostic tests, such as PD-L1 expression, can help identify patients who would benefit the most from ICIs and are widely used in various malignancies. However, this approach does not apply to HCC, where ICIs are administered to all patients regardless of the PD-L1 expression, knowing that some cases will be resistant to ICIs. For example, in the atezolizumab/bevacizumab arm of the landmark IMbrave 150 study, 19% of the patients were refractory to treatment [51]. This highlights the need for predictive markers that can help select patients who are most likely to benefit from ICIs. This is particularly relevant in the context of the graft rejection risk when utilizing ICIs as bridging treatment to liver transplant.

##### PD-L1 Expression

In the Checkmate-040 trial, which compared nivolumab monotherapy to combination therapy with ipilimumab in advanced HCC patients who previously received sorafenib, an objective response was observed regardless of the PD-L1 status [52]. Similarly, in the HIMALAYA trial, treatment with durvalumab and tremelimumab showed OS benefit compared to durvalumab or sorafenib as single agents in advanced HCC regardless of the PD-L1 expression [53]. In the Checkmate-459 trial, which compared nivolumab vs. sorafenib for advanced HCC, patients with PD-L1 ≥1% achieved a higher response rate with nivolumab compared to those with PD-L1 expression <1% (28% vs. 12%). However, this did not translate into a survival benefit, and both subgroups had a similar median OS (16.1 months for PD-L1 ≥ 1% and 16.7 months for PD-L1 < 1%) [54]. The KEYNOTE-224 trial aimed to assess the safety and efficacy of pembrolizumab in patients with advanced HCC previously treated with sorafenib. The study found that PD-L1 expression assessed by a combined positive score (CPS) ≥1 was associated with response to pembrolizumab in a subgroup of patients. The tumor proportion score (TPS) ≥1% was not significantly associated with the response, which could imply that the combination of immune and tumor cell scoring might improve the predictive value of PD-L1 testing [55]. A systematic review and meta-analysis by Yang et al. evaluated the predictive value of PD-L1 in patients with HCC treated with ICIs, and it found that patients with a positive PD-L1 expression had better ORR (pooled odds ratio, 1.86, 95% CI, 1.35–2.55). However, there was no difference in the disease control rate compared to those who were PD-L1 negative [56].

##### Summary and Recommendations

In the absence of substantial evidence from clinical trials, PD-L1 expression is not recommended for routine use as a predictor of the response to ICI in advanced HCC. Taking into consideration that there is a significant inter-assay heterogeneity in detecting PD-L1 in HCC, as was evident in the Blueprint-HCC study [57], further efforts are needed to standardize the measurement of PD-L1 expression in HCC to improve its consistency as a potential biomarker of the response to ICIs.

##### Microsatellite Instability

Microsatellite instability (MSI) is the result of a deficiency in DNA mismatch repair (MMR) mechanisms. MSI-high tumors accumulate somatic mutations that lead to neoantigen formation, which in turn activates an immune inflammatory cascade, thus making these tumors sensitive to ICIs. The efficacy of ICIs in different types of MSI-high malignancies has been demonstrated in the KEYNOTE-016, -164, -012, -028, and -158 trials [58,59,60,61,62]. However, the prevalence of high MSI in HCC is low, and it was found to be between 0% and 2.9% in a review of the literature [63]. Despite the limited evidence supporting the use of immunotherapy in MSI-high HCC [64,65], the low occurrence of microsatellite instability in this cancer diminishes the value of the MMR status as a predictive biomarker.

##### Tumor Mutational Burden

The tumor mutational burden (TMB) refers to the number of DNA mutations per megabase (Mb) within the coding genome of tumors [66]. An elevated TMB (>10 mutations/Mb) leads to increased neoantigen expression, which can trigger an immune response, making it a potential predictive marker for the effectiveness of immunotherapy. A high TMB was significantly associated with better objective response rates (ORRs) with pembrolizumab in the Keynote-158 trial, which led to the FDA approval of this drug in advanced solid tumors with high TMB. However, the improved ORR did not translate into a survival benefit and none of the included patients with biliary cancers had a high TMB [67,68]. Indeed, compared to other types of malignancies, HCC has a relatively low TMB (a median of 5 mutations/Mb) [69]. Additionally, the TMB in HCC can fluctuate depending on the type of pathology sample between paraffin-embedded or fresh-frozen section, and on the geographic origin of the patient [66]. In summary, the methodological variations and low TMB in HCC render the use of this predictive tool limited.

##### Tumor-Infiltrating Lymphocytes

ICIs exert their anti-tumor effect by activating the immune cells in the tumor microenvironment. This includes tumor-infiltrating lymphocytes (TILs) such as cytotoxic CD8+ cells and natural killer (NK) cells. A post hoc analysis of the CheckMate 040 trial showed that increased CD3+ and CD8+ cell infiltration led to an improvement in the OS in patients treated with nivolumab, but it did not reach statistical significance [70].

##### Inflammatory Markers

Systemic inflammation, often triggered by a local pro-inflammatory response in the tumor microenvironment, can lead to poor survival outcomes in patients with cancer. Two inflammatory markers were studied in HCC patients treated with ICIs: the neutrophil to lymphocyte ratio (NLR) and the platelet to lymphocyte ratio (PLR). A study found that patients with HCC treated with nivolumab who achieved a partial or complete response had a significantly lower post-treatment NLR and PLR (*p* < 0.001 for both) compared to those with stable disease or progression. Both ratios were significantly associated with survival in a multivariable analysis [71]. In a retrospective analysis of 362 patients treated with ICIs for HCC, patients with NLR ≥ 5 had significantly worse OS (7.7 vs. 17.6 months, *p* < 0.0001), PFS (2.1 vs. 3.8 months, *p* = 0.025), and ORR (12% vs. 22%, *p* = 0.034). Patients with PLR ≥ 300 had similar results, with significantly shorter OS (6.4 vs. 16.5 months, *p* < 0.0001) and PFS (1.8 vs. 3.7 months, *p* = 0.0006) [66].

##### Summary and Recommendations

Although these results are encouraging, the utility of inflammatory markers as predictive biomarkers is limited by the fluctuation during the course of the illness and treatment. Further validation in larger cohorts is necessary.

##### Gut Microbiota

The commensal microbes of the digestive system are being highlighted as key players in cancer pathogenesis and the response to treatment in several types of malignancies, particularly in colorectal cancer [72]. A systematic review and meta-analysis assessed the relation of the gut microbiota composition and the response to ICIs in 775 patients with different types of solid organ malignancies. This effort resulted in identifying *Faecalibacterium prausnitzii*, *Streptococcus parasanguinis*, *Bacteroides caccae*, and *Prevotella copri* to be more commonly present in responders to ICIs and to be associated with a better prognosis. In contrast, *Blautia obeum* and *Bacteroides ovatus* were associated with a poorer prognosis [73]. Zheng et al. examined fecal samples from eight patients with HCC receiving camrelizumab and compared the microbiota of responders vs. non-responders at different time points during treatment. Responders were characterized by having higher taxa richness and more gene counts than non-responders. Furthermore, the study found a dynamic variation in the composition of the microbiota over the course of treatment, such as an increase in Proteobacteria in non-responders between weeks 3 and 12 of PD-1 blockade. Therefore, examining the variation in the gut microbiota can be a promising predictor of the response to ICIs in HCC as early as within 3 to 6 weeks of treatment administration [74]. In a larger cohort of 65 patients with advanced hepatobiliary cancers receiving anti-PD-1 treatment, *Lachnospiraceae* bacterium-GAM79 and *Alistipes* sp. Marseille-P5997 were significantly more enriched in responders, and their abundance was associated with better progression-free and overall survival. In contrast, the *Veillonellaceae* were significantly more abundant in non-responders and were associated with worse PFS and OS. Another observation from this study was that a more diverse gut microbiota is associated with a lower risk of immune-related adverse events [75]. Another study found that *Lachnoclostridium* enrichment conferred a survival benefit (median OS of 22.8 months vs. 5.6 months, *p* = 0.032), while patients with enriched fecal *Prevotella 9* had significantly worse OS compared to others (median OS of 8.6 months vs. 17.2 months, *p* = 0.039). The best median OS was in patients with both *Lachnoclostridium* enrichment and *Prevotella 9* depletion in the feces (22.8 months) [76].

##### Summary and Recommendations

Overall, the gut microbiota’s composition and its dynamic variation throughout immunotherapy administration is a promising biomarker to predict the response to treatment, but it still needs validation in larger studies. Additionally, its clinical application may be challenging given the high susceptibility of the microbiome to external factors, such as antibiotic administration and dietary changes, which may complicate the interpretation and utility of this marker.

##### Genomic Characteristics

HCC has a heterogeneous genomic profile, and several studies have been conducted to identify potential genomic biomarkers that can inform treatment decisions. HCCs can be categorized into two molecular types: proliferative and non-proliferative. Proliferative HCCs have a subclass characterized by a high number of infiltrating CD4+ and CD8+ T cells, which respond well to ICIs [77]. On the other hand, non-proliferative HCCs are dominated by Wnt signaling and tend to be less aggressive, have lower levels of AFP and are more differentiated [78]. The role of this molecular classification is not yet clear in clinical practice, and further evidence is needed to support its utility as a predictive biomarker.

A promising predictive biomarker is the Wnt-β-catenin pathway, since its activation is one of the main cancer-driver gene mutations in HCC, causing the upregulation of oncogenes and favoring immune resistance. Activation of this pathway occurs in 30% to 50% of cases of HCC, triggered by mutations in the CTNNB1 that encodes β-catenin and the inactivation of AXIN1 or APC, which are inhibitors of the Wnt pathway [11]. To test this in a clinical context, Harding et al. used next-generation sequencing to determine predictive and prognostic biomarkers for HCC. Among the patients treated with ICIs, activating alterations in the Wnt-β-catenin pathway was associated with worse outcomes in terms of the disease control rate (0% vs. 53%), median PFS (2.0 vs. 7.4 months), and median OS (9.1 vs. 15.2 months) [79]. Morita et al. evaluated the molecular and immunological features of HCC as predictive markers of the response to ICIs. The study concluded that the absence of staining of the molecules in Wnt/β-catenin signaling, high infiltration of CD8+ cells, and high CPS of PD-L1 were significant contributors to the response to PD-1 blockade [80].

##### Summary and Recommendations

Additional research in larger groups of patients is necessary to delineate the predictive value of HCC’s genomic characteristics and their correlation to treatments, including ICIs.

## 3. Future Directions

Multiple prospective studies are ongoing to assess the safety and efficacy of bridging systemic therapies before LT.

The PLENTY202001 trial is investigating the safety and efficacy of pembrolizumab in combination with lenvatinib as neoadjuvant therapy in 192 anticipated participants with HCC exceeding the Milan criteria before LT. The study aims to determine whether this combination as a neoadjuvant treatment for advanced HCC could decrease postoperative recurrence and to analyze potential immune biomarkers of the therapeutic response [81]. ESR-20-21010 is a single-arm, phase II, multicenter clinical trial aiming to assess the safety and efficacy of durvalumab and tremelimumab for the treatment of 30 anticipated patients with HCC within the UCSF criteria who are listed for liver transplant and have cirrhosis or portal hypertension. The primary endpoint of this study is post-transplant rejection within 30 days of transplant [82]. The combinations of ICIs plus either lenvatinib or bevacizumab are currently under investigation in two other clinical trials too [83,84]. Finally, Sun Yat-Sen Memorial Hospital in China is studying single-agent ICIs as bridging/downstaging treatment to LT in HCC patients who are beyond the Milan criteria [85].

### Summary and Recommendations

Ongoing clinical trials on systemic bridging therapies to LT are examining ICIs alone or in combination with either TKIs or bevacizumab (Table 4). An important endpoint is the rejection rates post LT. Publication of the results of these ongoing clinical trials will have a favorable impact on the management of this particular subset of HCC patients.

## 4. Conclusions

Enhancing HCC patients’ bridging to LT via systemic treatments is an evolving field, especially when locoregional liver-directed therapy is contraindicated. Investigations into single-agent TKIs, ICIs, or their combinations as bridging modalities are being conducted. ICIs in combination with TKIs or VEGF inhibitors likely represent the most promising approach. However, concerns remain regarding post-LT rejection after bridging with ICIs, necessitating further prospective clinical trials to determine the optimal pre-LT ICI washout periods to ensure safety and efficacy. Concurrently, efforts to identify biomarkers, particularly of ICIs, are underway to better predict the HCC tumor response to treatment and to mitigate the risks associated with post-LT rejection.

## Figures and Tables

**Figure 1 cancers-16-02081-f001:**
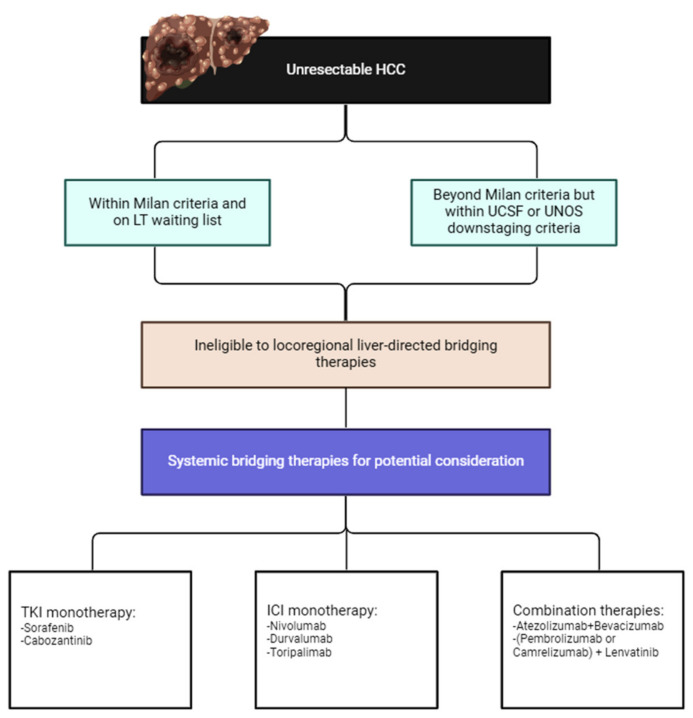
Potential role of systemic therapies as bridging treatments to LT in HCC.

**Table 1 cancers-16-02081-t001:** Overview of the studies on TKIs used as a bridging therapy to liver transplant.

Author/Year	Milan Criteria	Treatment	No. of Transplanted Patients	Post-LT DFS	Post-LT OS	Post-LT Follow-Up (Months)
Minoux et al. (2022) [21]	In: 69.4% Out: 30.7%	Sorafenib	26	48%	77%	60
Abdelrahim et al. (2022) [24]	In: 74% Out: 26%	TACE ± sorafenib	128	100% vs. 67.2%, *p* = 0.07	77.8% vs. 61.5%, *p* = 0.51	60
Bhardwaj et al. (2022) [26]	In: 100%	Cabozantinib	2	50%	50%	21
Golse et al. (2018) [20]	In: 60% Out: 40%	3/5: hepatectomy or TACE then sorafenib 2/5: sorafenib	5	100%	NA	27
Hoffmann et al. (2015) [23]	In: 100%	TACE + sorafenib vs. TACE + placebo	17	HR: 1.259 (95%CI: 0.486, 3.270)	NA	10
Kulik et al. (2014) [22]	up to UCSF criteria: 100%	Y90 radioembolization ± sorafenib	17	NA	72% vs. 70%, *p* = 0.57	36
Vitale et al. (2013) [19]	Out: 100%	Sorafenib	6	66%	66%	(27–41)
Saidi et al. (2010) [18]	In: 100%	Sorafenib	7	85%	NA	NA

Abbreviations: NA: not available, OS: overall survival, Post-LT: post-liver transplantation, TACE: transarterial chemoembolization, UCSF: University of California San Francisco.

**Table 2 cancers-16-02081-t002:** Overview of the studies on ICIs used as a bridging therapy to liver transplant.

Author/Year	Milan Criteria	Treatment	No. of Transplanted Patients	Washout Period (Days)	Post-LT IS-Protocol Included Steroids	BPAR	HCC Recurrence	Post-LT Follow-Up (Months)
Schmiderer et al. (2023) [42]	Out: 100%	Atezolizumab + bevacizumab	1	42	Yes	No	No	12
Abdelrahim et al. (2022) [41]	Out: 100%	Atezolizumab + bevacizumab	1	60	No	No	No	12
Dave et al. (2022) [43]	In: 87% Out: 13%	Nivolumab	5/8	11–354 (median: 105)	NA	Yes: 40%	No	NA
Schnickel et al. (2022) [35]	NA	Nivolumab	5	10–330	NA: 40% No: 20% Yes: 40%	Yes: 40%	No	2–16
Tabrizian et al. (2021) [33]	up to UCSF criteria: 100%	Nivolumab	9	1–253 (80% of patients ≤ 30 days)	Yes	No	No	8–23 (median: 16)
Chen et al. (2021) [34]	Out: 100%	Nivolumab	5	Mean: 63.80 ± 18.26	No	No	Yes: 40%	NA
Sogbe et al. (2021) [36]	Out: 100%	Durvalumab	1	90	Yes	No	No	24
Chen et al. (2021) [37]	In: 100%	Toripalimab	1	93	Yes	Yes: 100%, fatal hepatic necrosis	NA	NA
Qiao et al. (2021) [40]	NA	(Pembrolizumab or camrelizumab) + lenvatinib	7	42	Yes	Yes: 14.3%	NA	NA
Nordness et al. (2020) [39]	In: 100%	Nivolumab	1	8	Yes	Yes: 100%, fatal hepatic necrosis	NA	NA
Schwacha-Eipper et al. (2020) [38]	In: 100%	Nivolumab	1	105	NA	No	No	12

Abbreviations: BPAR: biopsy-proven acute rejection, IS: immunosuppression, NA: not available, Post-LT: post-liver transplantation, UCSF: University of California San Francisco.

**Table 3 cancers-16-02081-t003:** The RECIST and mRECIST criteria for response assessment in HCC.

RECIST Criteria	mRECIST Criteria
CR: disappearance of all target lesions.	CR: disappearance of any intra-tumoral arterial enhancement in all target lesions.
PR: ≥30% reduction of the sum of the diameters of target lesions.	PR: ≥30% reduction of the sum of the diameters of viable (enhancing) target lesions.
SD: features classified as neither PR nor PD.	SD: features classified as neither PR nor PD.
PD: ≥20% increase of the sum of the diameter of target legions.	PD: ≥20% increase of the sum of the diameter of viable (enhancing) target legions.

Abbreviations: CR: complete response; PR: partial response; SD: stable disease; PD: progressive disease.

**Table 4 cancers-16-02081-t004:** Overview of the ongoing studies on systemic bridging therapy to liver transplant.

ClinicalTrials.gov ID	Bridging Therapy	Trial Phase	Primary Endpoints
NCT04425226 [81]	Pembrolizumab + lenvatinib	NA	RFS
NCT05027425 [82]	Durvalumab + tremelimumab	2	Cellular rejection rate
NCT05185505 [83]	Atezolizumab + bevacizumab	4	Acute rejection rate post liver transplant
NCT04443322 [84]	Durvalumab + lenvatinib	NA	RFS, PFS
NCT05475613 [85]	Anti-PD-1 inhibitors (tislelizumab, pembrolizumab, nivolumab)	2	2-year event-free survival rate

Abbreviations: ID: identifier, NA: not available, PFS: progression-free survival, PD-1: programmed cell death protein 1, RFS: recurrence-free survival.

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
