# Peer review of "Hepatocellular Carcinoma: The Evolving Role of Systemic Therapies as a Bridging Treatment to Liver Transplantation"

_cancers, 2024, doi:10.3390/cancers16112081_

Round 1

Reviewer 1 Report

Comments and Suggestions for Authors

Interesting and well written article.

The authors should comment more on the potential role and rationale of use of lenvatinib in this setting, based also on comparative data in HCC patients (cite the recent SRMA: PMID: 34017396)

Some figures would help the quality of the manuscript

I don't understand the need for the section on summary and recommendations after each section....

The section on inflammatory markers should be expanded as we have more data in the literature

Author Response

Dear reviewer,

Thanks for your comments, please find our responses below:

  1. We have added a new paragraph about the potential role of Lenvatinib in the TKI (section 2.1). The paragraph is highlighted in yellow and new references are included in brackets. It is worth noting that we commented on Lenvatinib role in combination with immunotherapy in section 2.2.1. We, also, commented on the potential role of Lenvatinib in the ongoing trials in section 3.
  2. We have created a figure in the introduction (section 1) illustrating the potential role of systemic therapies as bridging treatments to liver transplant in HCC.
  3. The (summary and recommendations) after each section are intended to present the information concisely to the readers. We have aimed to facilitate the reading experience after each section.
  4. The section on inflammatory markers is brief as there is no practice-changing data available on this topic, and it is not a primary focus of the review.

Reviewer 2 Report

Comments and Suggestions for Authors

Manuscript ID: cancers-3017957

Type of manuscript: Review

Title: HCC: Evolving Role of Systemic Therapies as a Bridging Treatment to Liver Transplantation

Authors: Yacob Saleh et al.

This was an informative review about systematic chemotherapy for inoperative HCC. However, there was a little weak impression as review of the Bridging Treatment to Liver Transplantation. 

The chapter of 2.2.3. Biomarkers was a review of predictors for ICIs treatment, not bridging treatment to LT. 

Next, now ICI monotherapy, combination with ICI and TKI, or combination with ICI and ICI are available for inoperative HCC. Were there any differences in rejection rate in post-ICI LT among three regimens? 

Author Response

Dear reviewer,

Thanks for your comments, please find our responses below:

  1. The review is not intended to discuss systemic chemotherapy for inoperative HCC. The review addresses the evolving role of systemic therapies as a bridging treatment to liver transplantation in HCC.
  2. We have added a section on biomarkers as identifying biomarkers, particularly to immunotherapy, will better predict HCC tumor response to treatment and will mitigate risks associated with post-liver transplant rejection as discussed in the conclusion section.
  3. There are no direct comparisons on rejection rates. However, retrospective data showed that immunotherapy doublet has higher rejection rate than monotherapy but these are retrospective in nature and has small sample number.

Round 2

Reviewer 1 Report

Comments and Suggestions for Authors

The authors consistently improved the paper.

I have just one more remark: the added references should be correctly listed in the bibliography and not put as PMID/doi in the text.

Author Response

Dear Reviewer,

Thanks  for your comments.

We have updated all references numbers through the article and all references are correctly listed in the bibliography.

Best regards,

Yacob Saleh

Round 3

Reviewer 1 Report

Comments and Suggestions for Authors

The revised manuscript is OK. Thank you!